# Metabolomic and transcriptomic signatures of prenatal excessive methionine support nature rather than nurture in schizophrenia pathogenesis

Siwei Chen[1,2,5], Wedad Alhassen[3,5], Ryan Yoshimura[3], Angele De Silva [4], Geoffrey W. Abbott[4], Pierre Baldi[1,2] & Amal Alachkar [2,3 ✉]

The imbalance of prenatal micronutrients may perturb one-carbon (C1) metabolism and increase the risk for neuropsychiatric disorders. Prenatal excessive methionine (MET) produces in mice behavioral phenotypes reminiscent of human schizophrenia. Whether in-utero programming or early life caregiving mediate these effects is, however, unknown. Here, we show that the behavioral deficits of MET are independent of the early life mother-infant interaction. We also show that MET produces in early life profound changes in the brain C1 pathway components as well as glutamate transmission, mitochondrial function, and lipid metabolism. Bioinformatics analysis integrating metabolomics and transcriptomic data reveal dysregulations of glutamate transmission and lipid metabolism, and identify perturbed pathways of methylation and redox reactions. Our transcriptomics Linkage analysis of MET mice and schizophrenia subjects reveals master genes involved in inflammation and myelination. Finally, we identify potential metabolites as early biomarkers for neurodevelopmental defects and suggest therapeutic targets for schizophrenia.

[1] Department of Computer Science, School of Information and Computer Sciences, University of California-Irvine, Irvine, CA 92697, USA. [2] Institute for Genomics and Bioinformatics, School of Information and Computer Sciences, University of California-Irvine, Irvine, CA 92697, USA. [3] Department of Pharmaceutical Sciences, University of California-Irvine, Irvine, CA 92697, USA. [4] Bioelectricity Laboratory, Department of Physiology and Biophysics, School of Medicine, University of California-Irvine, Irvine, CA 92697, USA. [5] These authors contributed equally: Siwei Chen, Wedad Alhassen. ✉email: aalachka@uci.edu

The necessity for a finely tuned one-carbon (C1) metabolism activity during pregnancy is supported by the role of micronutrients such as methionine, choline, vitamin B12, betaine, and folate in brain development through involvement in methylation processes[1–9]. C1 metabolism encompasses two complex pathways, the transmethylation (methionine) and the folate cycles, through which a carbon unit is transferred from one to the other metabolic pathways. Dietary or supplementary methionine is converted to S-adenosyl-methionine (SAM), the main methyl donor that is used in almost all methylation reactions, where it is converted to S-adenosyl-homocysteine (SAH) (Fig. 2d). SAH is converted to homocysteine (Hcy), a potent neurotoxin, which can then be recycled to methionine by receiving methyl group provided by the folate cycle. Methylation mediated through the C1 pathways is a universal reaction that plays a critical role in numerous biological processes and metabolic pathways that are involved in cell proliferation, differentiation, survival, and other cellular functions[10–14].

Fluctuations in the methyl-donor dietary intake during pregnancy have been shown to cause changes in the components of the C1 pathways and in the levels of DNA methylation[15].

The imbalance of methyl-donor micronutrients during gestational and development stages may lead to perturbations of C1 metabolism. C1 metabolism impairments increase the risk for the development of neuropsychiatric deficit such as cognitive deficits, schizophrenia and autism spectrum disorders (ASD), and produces in animals permanent phenotypic changes that resemble those in the major neuropsychiatric disorders[16–22]. We recently examined the effects that disruption of C1 metabolism during gestation have on mice progeny. Pregnant mice were administered methionine equivalent to twice their daily dietary intake in their final week of gestation at the time of brain development[23]. The resulting pups exhibited a constellation of behavioral, electrophysiological and transcriptomic changes consistent with schizophrenia. For example, MET mice displayed social deficits, cognitive impairments and augmented stereotypy, accompanied with decreased synaptic plasticity, and reduced local excitatory synaptic connections in the hippocampus. The schizophrenia-related genes Npas4, Arc, and Fos were downregulated in the MET mice. Perturbation of C1 metabolism micronutrients during pregnancy has important epidemiological and therapeutic implications because of the exceptionally high potential impact on prevention or therapeutic intervention of psychiatric disorders. Therefore, the aim of this study is to investigate the early changes in the brain metabolome and transcriptome linked to methionine overload during the third week of pregnancy in mice, and whether the behavioral phenotype induced by excessive prenatal methionine can be rescued by caregiving during early life stages. The timing of the MET administration in the third week of mouse pregnancy was selected to correspond to the comparable stage of human pregnancy. Given the alterations in methyl-donor requirements and methionine metabolism during the different gestation stages, a higher rate of transmethylation occurs in the 3rd trimester of gestation in pregnant women to meet the increase in methylation demands[24]. Further, maternal hyperhomocysteinemia (as a result of the disrupted methionine metabolism) during the third trimester of pregnancy has been be associated with a two-fold increase in the risk of schizophrenia in the adult offspring[22].

## Results

### MET mice display normal maternal behavior.
The latencies to retrieve the first pups by SAL and MET-treated mice were comparable on PPD2, $P > 0.05$ (Fig. 1a). Similarly, the retrieval durations (total time to retrieve three pups) were not significantly different between the two groups $P > 0.05$ (Fig. 1b), and dams did not differ in their nest building (Fig. 1c). These results indicate that MET administration during gestational stage does not affect maternal behaviors.

### Cross-fostering did not change the behavioral deficits induced by prenatal MET.
Animals born to a MET female and reared by a SAL female (MET → SAL) displayed similar behavioral phenotypes to those born to- and raised by their MET biological mothers (MET → MET): the MET → SAL displayed increased locomotor activity ($P < 0.01$, one-way ANOVA followed by Tukey's post test, Fig. 1e), decreased PPI ratios in the sensorimotor gating behavior ($P < 0.001$, one-way ANOVA followed by multiple comparisons test, Fig. 1f), and decreased percentage of correct arm choice alternation in the spontaneous T-maze alternation assay ($P < 0.001$, one-way ANOVA followed by Tukey's post test, Fig. 1g) compared to SAL-SAL. Similarly, mice born to a SAL female and reared by a MET female (SAL → MET) displayed similar behavioral phenotypes to those born to- and raised by their SAL biological mothers (SAL → SAL) (Fig. 1e–g).

### Targeted metabolites profiling.
The global brain metabolomics analysis of 24-h-old pups reveals alterations in the MET pups' brain in the components of C1 metabolism reflected by the decrease in methionine sulfoxide, and increase in SAM, betaine, glycine, glutathione, methylglycinate, and serine (Fig. 2a–d, Supplementary Table 1). There were profound lipidomic abnormalities including membrane, energy, and signaling lipids. The lipidomic abnormalities can be summarized by the increase in ceramide, phosphatidylcholines (PC), lysophosphatidylcholines (LPCs), lysophosphatidylethanolamines (LPE), saturated and unsaturated fatty acids (FA), arachidonic acid, glucoceramides (GlcCer), Phosphtatidylglycerols (PG 34:2; 16:1-18:1), and the decrease in sphingomyelins (SMs), diacylglycerols (DG34:1), and Phosphtatidylglycerols (PG 44:12; PG 22:6) (Fig. 2a–c, e, Supplementary Table 1). Mitochondrial lipid metabolism was also altered by prenatal methionine, evidenced by the 17-fold increase in malonyl-carnitine, in the brains of 24-h-old MET pups (Fig. 2a,b, Supplementary Table 1). Finally, prenatal methionine produced changes in the levels of amino acids' neurotransmitters (decreased glutamate and glutamine, and increased aspartate) and the neurotransmitters' precursors (decreased tryptophan and tyrosine) (Fig. 2a–c, Supplementary Table 1).

Analysis of the metabolomics pathways by Metaboanalyst revealed profound dysregulations of several pathways, on top of which is purine pathway (Fig. 2f, g, Supplementary Table 2), but also include mitochondrial electron transport chain, and fatty acid elongation in mitochondria (Fig. 2f, h, Supplementary Table 2).

In the adult brains, among the metabolites whose levels changed substantially are several lipids, guanosine, N1-Methyladenosine, and UDP-N-acetylglucosamine, (Fig. 2i, Supplementary Table 3). Analysis of the metabolomics pathways by Metaboanalyst revealed dysregulation in the brains of MET adult pups of the oxidation of Branched Chain Fatty Acids ($P < 0.05$, Fig. 2j, Supplementary Table 4).

### Transcriptomics analysis.
The global brain transcriptomic analysis was carried out using mRNA microarray analysis covering 28,853 genes to determine which genes' expression are affected in the MET mice (Fig. 3a–c, Supplementary data 1). We found 795 genes displaying changes in expression (with a ≥ 1.2-fold change, shown in Supplementary data 1). Among the subset of genes that exhibited $a \geq 1.5$-fold change ($P < 0.05$), 20 genes were

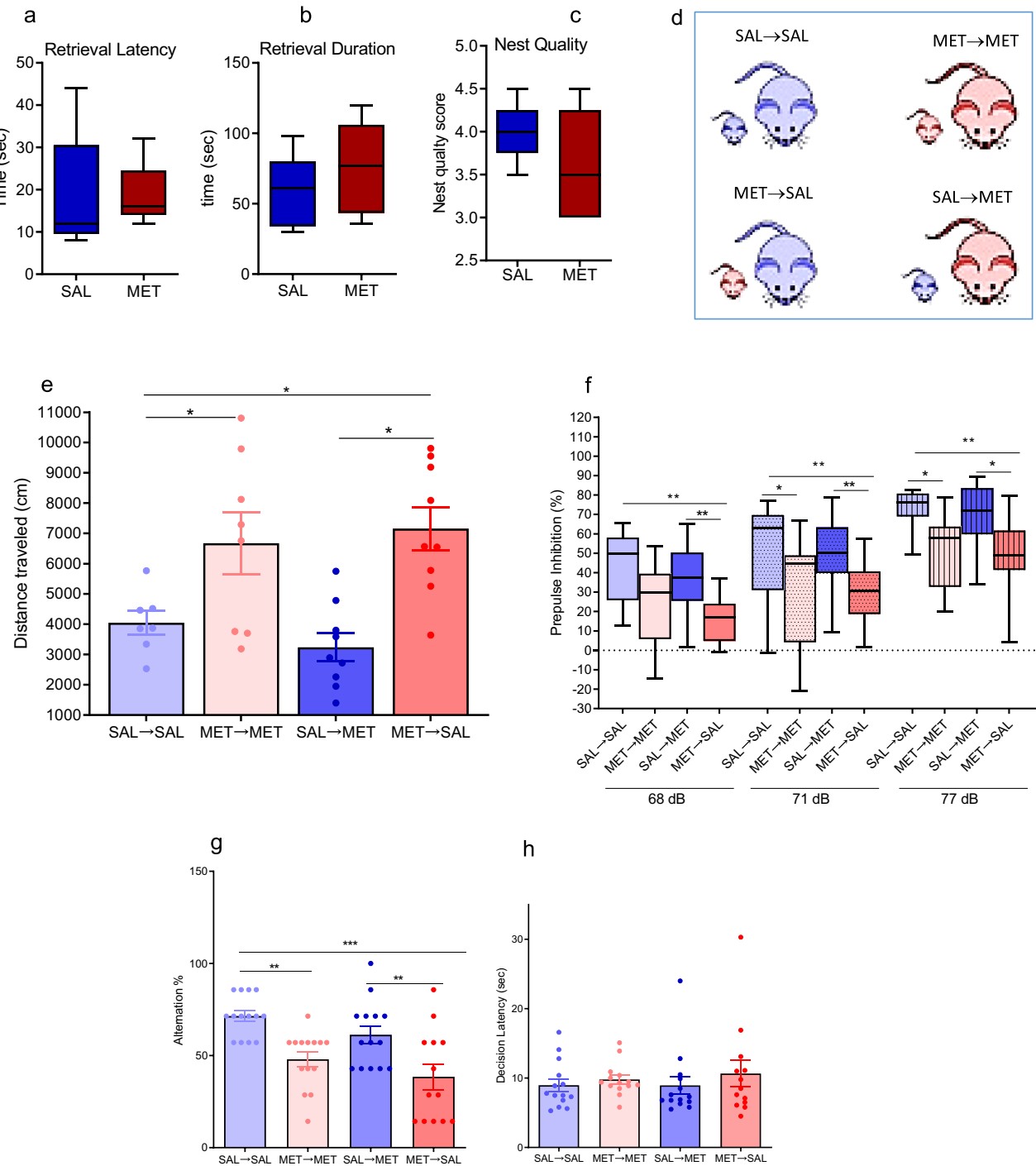

**Fig. 1 Maternal behavior is normal in MET mice and cross-fostering does not affect behavioral phenotype in MET mice. a–c** Maternal behaviors assessed in postpartum mice injected with SAL or MET during their last week of pregnancy. Box-whisker plots show median values and 25th/75th percentiles with minimum and maximum whiskers. **a** Pups retrieval latency by SAL and MET mothers, ($n_{SAL} = 5$, $n_{MET} = 5$), unpaired $t$-test, $P > 0.05$. **b** Pups retrieval duration of SAL and MET mice, ($n_{SAL} = 5$, $n_{MET} = 5$), unpaired t-test, $P > 0.05$. **c** The score of the nesting quality of SAL and MET mice, ($n_{SAL} = 5$, $n_{MET} = 5$), unpaired $t$-test, $P > 0.05$. **d** Diagram showing the design of cross-fostering experiment. **e–h** Behavioral tests assessed in adult male pups. **e** Distance mice travelled in 60 min of the locomotion assay ($n_{SAL-SAL} = 7$, $n_{MET-MET} = 8$, $n_{SAL-MET} = 9$, $n_{MET-SAL} = 9$), One-way ANOVA test ($F = 4.8$, $P = 0.0058$): *$P < 0.05$. Data are presented as means ± S.E.M. **f** Prepulse inhibition ratios against three prepulse stimulations in the PPI assay ($n_{SAL-SAL} = 11$, $n_{MET-MET} = 13$, $n_{SAL-MET} = 14$, $n_{MET-SAL} = 13$). One-way ANOVA followed by multiple comparisons test ($F = 14.26$, $P < 0.0001$), *$P < 0.05$, **$P < 0.01$. Box-whisker plots show median values and 25th/75th percentiles with minimum and maximum whiskers. **g** Percentage of the alternation choice mice made in the T-maze spontaneous assay ($n_{SAL-SAL} = 14$, $n_{MET-MET} = 14$, $n_{SAL-MET} = 14$, $n_{MET-SAL} = 13$). One-way ANOVA followed by Tukey post-test ($F = 8.8$, $P = 0.0002$), *$P < 0.05$, **$P < 0.01$, ***$P < 0.001$. Data are presented as means ± S.E.M. **h** Decision latency in the T-maze spontaneous assay. One-way ANOVA followed by Tukey post-test ($F = 3.667$, $P > 0.05$).

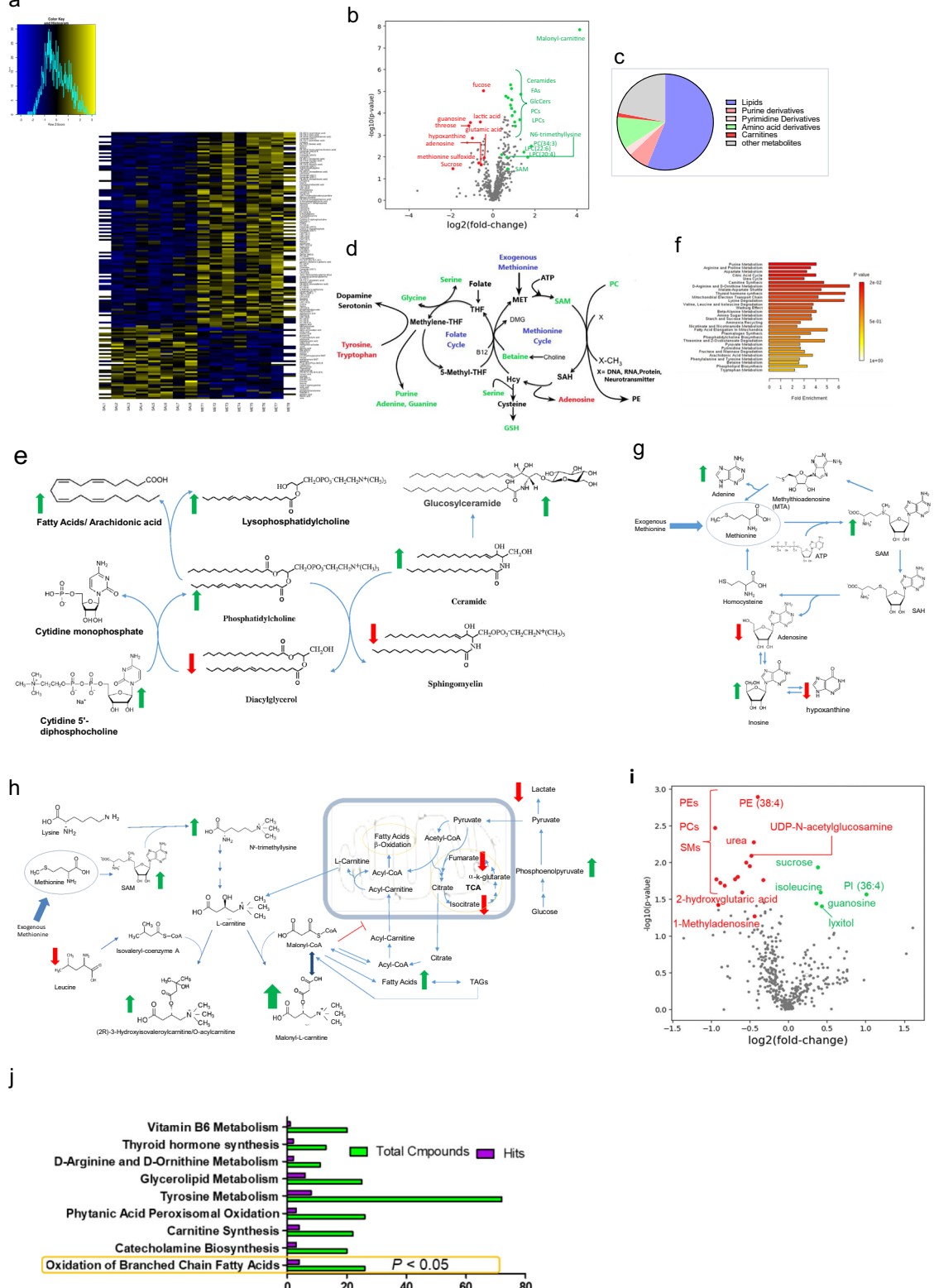

**Fig. 2 Prenatal MET causes changes in brain metabolites. a** Heatmap of metabolites significantly changed in the brains of MET newborn pups ($P < 0.05$).
**b** Volcano plot of metabolites significantly changed in the brains of MET newborn pups ($P < 0.05$). **c** Venn diagram of differentially expressed metabolites
showing the metabolite subfamilies. **d** Effect of prenatal MET on the one-carbon metabolism. Increased metabolites in green, and decreased metabolites in
red. **e** Changes of metabolites related lipid synthesis/metabolism. Increased metabolites in green, and decreased metabolites in red. **f** Metaboanalyst
pathway analysis of metabolites substantially changed in the newborn MET pups (Supplementary Table 2 shows details of number of compounds, hits and
*p* values). **g–h** changes of metabolites related to **g** purine metabolism, **h** carnitine synthesis and citric acid cycle. Increased metabolites in green, and
decreased metabolites in red. **i** Volcano plot showing differentially expressed metabolites in brains from adult MET versus SAL mice. **j** Metaboanalyst
pathway of metabolites substantially changed in the adult MET pups (Supplementary Table 4 shows details of number of compounds, hits and *p* values).

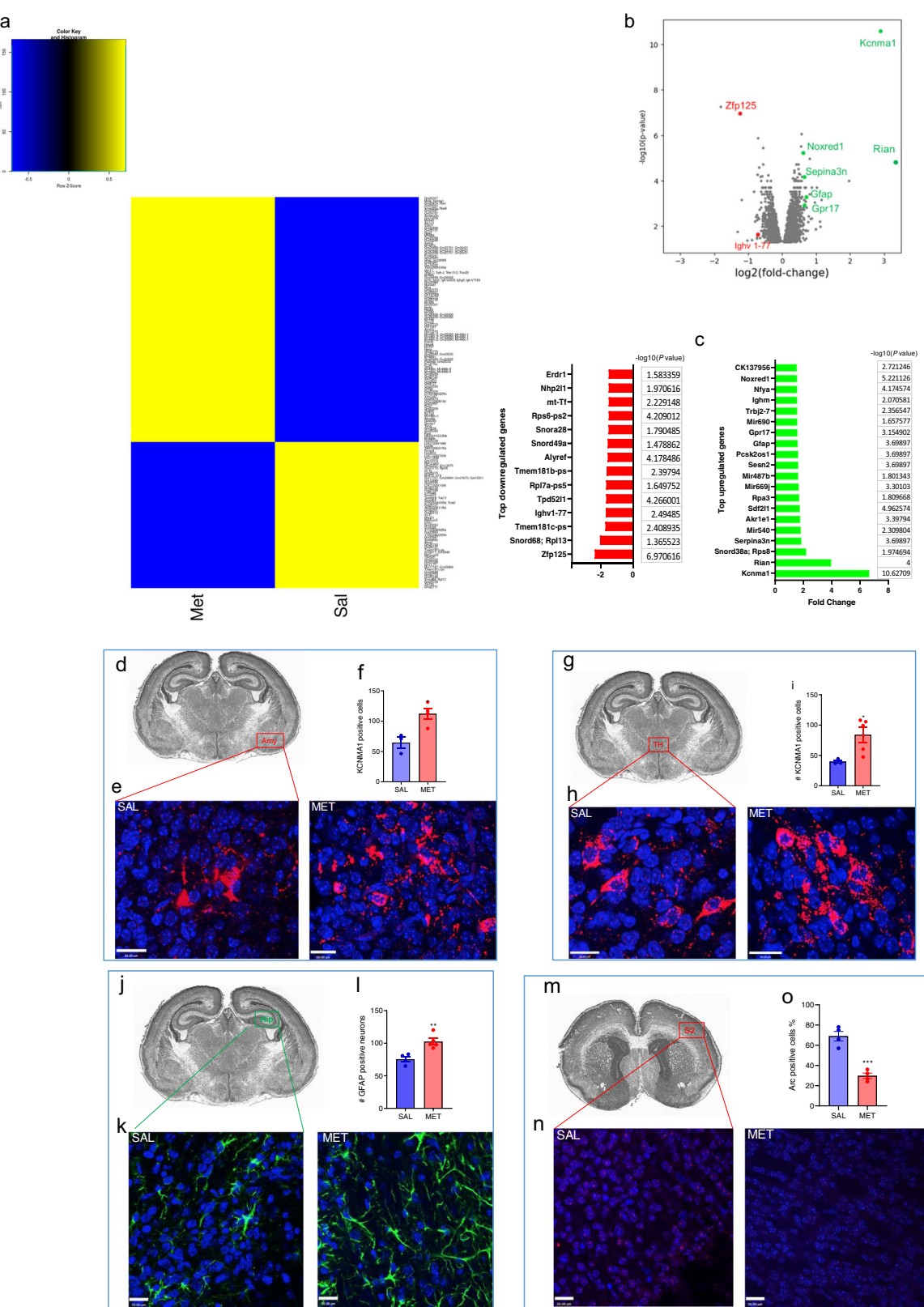

upregulated: Kcnma1, Rian, Rps8, Snord38a; Serpina3n, Akr1e1, Sdf2l1, Rpa3, Sesn2, Gfap, Gpr17, Mir540, Mir669j, Mir690, Pcsk2os1, Trbj2-7, Ighm, Nfya, Noxred1, and CK137956; and 14 genes were downregulated: Zfp125, Rpl13 (Snord68), Tmem181c-ps, Ighv1-77, Tpd52l1, Rpl7a-ps5, Tmem181b-ps, Alyref, Rps6-ps2, mt-Tf, Nhp2l1, and Erdr1. Using arbitrary fold change (FC) cutoffs of ≥5 and significance P-values of 0.01, only

Kcnma1 showed a statistically significant change (an increase by 6.63-fold in MET versus SAL offspring, $P = 2.36E-11$) in whole brain tissues (Supplementary data 1). Nine RBPs and 51TFs were among the genes that were differentially expressed in the brains of MET mice (Supplementary Table 5). Pathway enrichment analysis revealed dysregulation of several lipid pathways, including cholesterol biosynthesis, steroids metabolism, Sterol regulatory

**Fig. 3 Prenatal methionine treatment affects genes involved in neuronal activity, lipid metabolism, and neuro-inflammation. a** Heatmap analysis of differentially expressed genes ($P < 0.05$) in brains from MET versus SAL mice. **b** Volcano plot showing differentially expressed genes in brains from MET versus SAL mice. Kcnma1 exhibited the largest expression change among the upregulated genes (green symbols); Zfp125 exhibited the largest expression change among the downregulated genes (red symbols). **c** List of top downregulated (left) and upregulated genes in the brains of MET pups ($\geq$1.5-fold change, $P < 0.05$, with their $-\log10$(p-value). **d–i** Immunohistochemistry analysis of Kcnma1 in the brains of newborn SAL and MET pups. **d** Location of the Kcnma1 immunostaining in the amygdala on a nissl stain of a transverse brain section from a newborn mouse (Image Credit: Mouse Brain Atlas[85]), **e** representative image of Kcnma1 -immunoreactivity (red fluorescence) in the amygdala, **f** number of Kcnma1-immunoreactive cells in the amygdala in SAL and MET pups, unpaired t-test, $t = 3.61 P = 0.015$, $n = 3$-5; **g** location of the Kcnma1 immunostaining in the thalamus on a nissl stain of a transverse brain section from a newborn mouse (Image Credit: Mouse Brain Atlas), **h** representative image of Kcnma1-immunoreactivity in the thalamus, **i** number of Kcnma1-positive cells in the thalamus number of Kcnma1-immunoreactive cells in the amygdala in SAL and MET pups, unpaired t-test, $t = 2.92$, $P = 0.026$, $n = 3$-5. Scale bar $= 20\,\mu M$; Values represent mean $\pm$ SEM. **j–l** Immunohistochemistry analysis of GFAP in the brains of newborn SAL and MET pups. **j** Location of the GFAP immunostaining in the hippocampus on a nissl stain of a transverse brain section from a newborn mouse (Image Credit: Mouse Brain Atlas), **k** representative image of GFAP-immunoreactivity (green fluorescence) in the hippocampus, **l** number of GFAP-immunoreactive cells in the hippocampus in SAL and MET pups, Scale bar $= 20\,\mu M$, Values represent mean $\pm$ SEM and were analyzed by unpaired t-test, $t = 4.16$, $P = 0.0042$, $n = 3$-5. **m–o** Immunohistochemistry analysis of Arc in the sensory cortex (S2) of the newborn SAL and MET pups. **m**. location of the Arc immunostaining in the hippocampus on a nissl stain of a transverse brain section from a newborn mouse (Image Credit: Mouse Brain Atlas); **n** representative image of Arc-immunoreactivity (red fluorescence) in the sensory cortex S2; **o** quantification of Arc immuno-positive cells in the cortex, unpaired t-test, $t = 7.2$, $P = 0.0004$, $n = 3$-5). Scale bar $= 20\,\mu M$. Values represent mean $\pm$ SEM.

element-binding proteins (SREBP) signaling, fatty acid metabolism, carnitine acyl pathways (Supplementary data 2).

### MET pups exhibit increased Kcnma1 and Gfap and decreased Arc staining.

Results of immunohistochemistry using KCNMA1 antibody revealed an increase in the number of cells that express Kcnma1 ($P < 0.05$, Fig. 3d–i) in the amygdala and thalamus. The results reveal > twofold increase in Kcnma1 staining in the MET mice amygdala ($P < 0.05$, Fig. 3d–f) and thalamus ($P < 0.05$, Fig. 3g–i). We also found that the number of cells expressing GFAP in the hippocampus was increased in the MET group in comparison to the saline group ($P < 0.05$, unpaired t-test, Fig. 3j–l). Arc is a protein known to be involved in synaptic plasticity. We found that in the MET newborn pups, there is lower number of Arc-positive cells in the somatosensory cortex compared to the SAL group ($P < 0.05$, unpaired t-test, Fig. 3m–o). We found that Arc protein expression in both SAL and MET newborn pups is majorly contained in the cell nucleus (Fig. 3n).

### Integration of transcriptomic and metabolomics.

The linkage analysis of the changes in gene expression levels with the changes in metabolite levels in the newborn pups revealed master pathways including glutamate neurotransmission, SAM mediated methylation of DNA and RNA, mitochondrial NAD+ mediated oxidation-reductions, glutathione metabolism, and lipid synthesis and metabolism (Fig. 4a–f, Supplementary data 3). The increases in LPCs, LPE, fatty acids, and arachidonic acid were associated with overexpression of phospholipase enzymes such as Pla2g4b, Pla2g6, Plcxd2, Plcb3, and Napepld which hydrolyze phospholipids (Supplementary data 1). The linkage analysis of the changes in gene expression levels with the changes in metabolite levels in the adult pups revealed master pathways including lipid metabolism (Supplementary data 3). Increased SAM levels were associated with an increase in the expression of several genes encoding methylation enzymes that recruit SAM as a methyl-donor, including the DNA, mRNA, rRNA, and tRNA methylation enzymes: Mettl4, Mettl3, Tfb1m, and Trmt44 respectively (Fig. 4d, Supplementary data 3). The decrease of glutamate levels in MET pups was associated with the dysregulations of 17 genes involved in glutamate neurotransmission including glutamate transport (Slc16a1, Slc1a2/ Slc1a4, Slc7a1), synthesis (Gls2, Got1l1, Gpt, Gpt2, Bcat1, and Bcat2), release (Kcnma1), receptors (Grin2a, Grin2d, Grm1, Grm7, and Grm8), and the regulation of NMDA receptors (Kcnma1) (Fig. 4f, Supplementary Data 3).

### Integration of mouse-human transcriptomics.

Metabolite set enrichment analysis of the metabolites in the newborn MET mice using Metaboanalyst revealed statistically significant association with schizophrenia, Rett syndrome, and Alzheimer's among the top associated diseases (Fig. 5a, Supplementary Table 6). The linkage analysis of the transcriptomics of the newborn MET mice with the gene expression changes in schizophrenia patients revealed 552 common genes. Among the top genes whose expression showed the largest fold change are Serpina3n (human serpina3), Sesn2, Gfap, and Gpr17, (Fig. 5b, Supplementary Data 4). The linkage analysis of the transcriptomics of the newborn MET mice with the gene expression changes in autism patients revealed 381 common genes. Among the top genes, whose expression showed the largest fold change are Serpina3n, Gfap, Rpa3, Tpd52l1, and Sesn2 (Supplementary Fig. 1a, Supplementary Data 4). The same analysis revealed 197 common genes among newborn MET mice, schizophrenia, and autism (Supplementary Data 4), with Serpina3n being the gene that showed the largest fold change in the three groups.

The linkage analysis of the transcriptomics of the adult MET mice with the gene expression changes in schizophrenia patients revealed 315 common genes. The top genes whose expression showed the largest fold change included Npas4, Arc, Dusp1, Fos, Egr2, and Uck2 (Fig. 5c, Supplementary Data 4). The linkage analysis of the transcriptomics of the adult MET mice with the gene expression changes in autism patients revealed 217 common genes. Among the top genes whose expression showed the largest fold change included Tsnax, Rrl13a, Slc11a1, Mc4r, and Plscr4 (Supplementary Fig. 1b, Supplementary Data 4). The same analysis revealed 113 common genes among adult MET mice, schizophrenia, and autism (Supplementary Data 4).

### Discussion

In the present study, we demonstrated that the long-lasting behavioral deficits induced by excessive prenatal methionine administration, which mimic schizophrenia-like symptoms, are due to in-utero aberrations rather than through early life mother-infant interaction. We identified brain metabolites and transcriptomic signatures in the early life associated with the schizophrenia-like behavioral phenotype, which can serve as early biomarkers and therapeutic targets.

Given that administering high doses of methionine to adult male mice induces schizophrenia-like symptoms[25], and that methionine administration to schizophrenia patients exacerbates their psychotic symptoms[26–29], we asked whether the behavioral

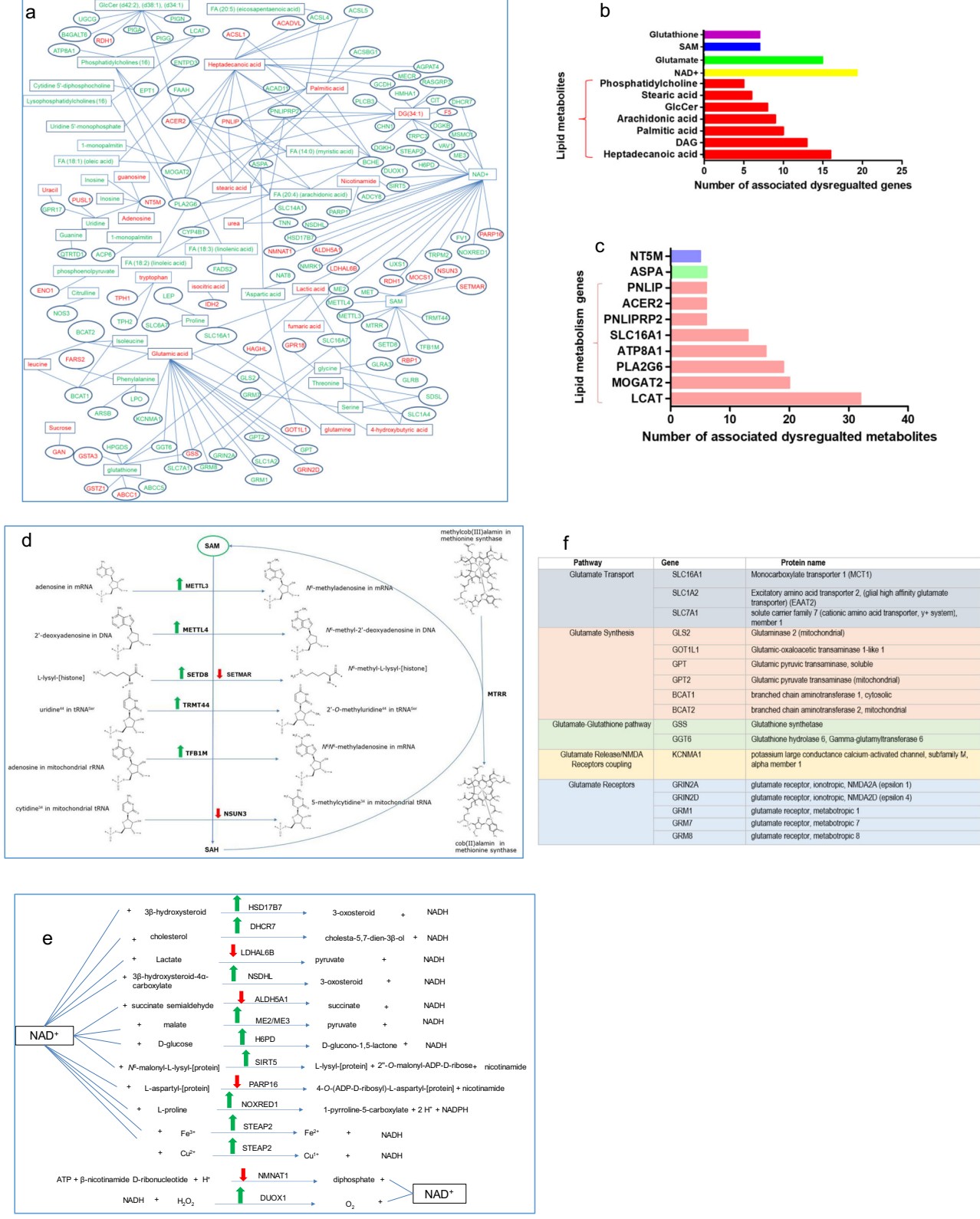

**Fig. 4 Integrated metabolite-transcriptomic analysis reveals abnormal pathways of lipids, glutamate, glutathione, and NAD+. a** Integrated network of functional interactions between genes and metabolites in 24-old MET brains. Metabolites are in rectangles, genes are in ovals, significantly increased metabolites and gene expressions in green, significantly decreased metabolites and gene expressions in red (*P* < 0.05). **b** List of metabolites whose expressions were significantly changed (*P* < 0.05) in MET pups and were associated with the largest gene networks (≥5 gene/metabolite) **c** List of genes whose expressions were significantly changed (*P* < 0.05) in MET pups and were associated with the largest metabolite networks (≥5 metabolite/gene). **d** SAM-associated pathways involving genes whose levels were significantly changed (*P* < 0.05) in MET pups, using UniPort for identifying the metabolic reactions[86]. **e** NAD+-associated pathways involving genes whose levels were significantly changed in MET pups (*P* < 0.05) using UniPort for identifying the metabolic reactions[86]. **f** Glutamate-associated pathways involving genes whose levels were significantly changed (*P* < 0.05) in MET pups.

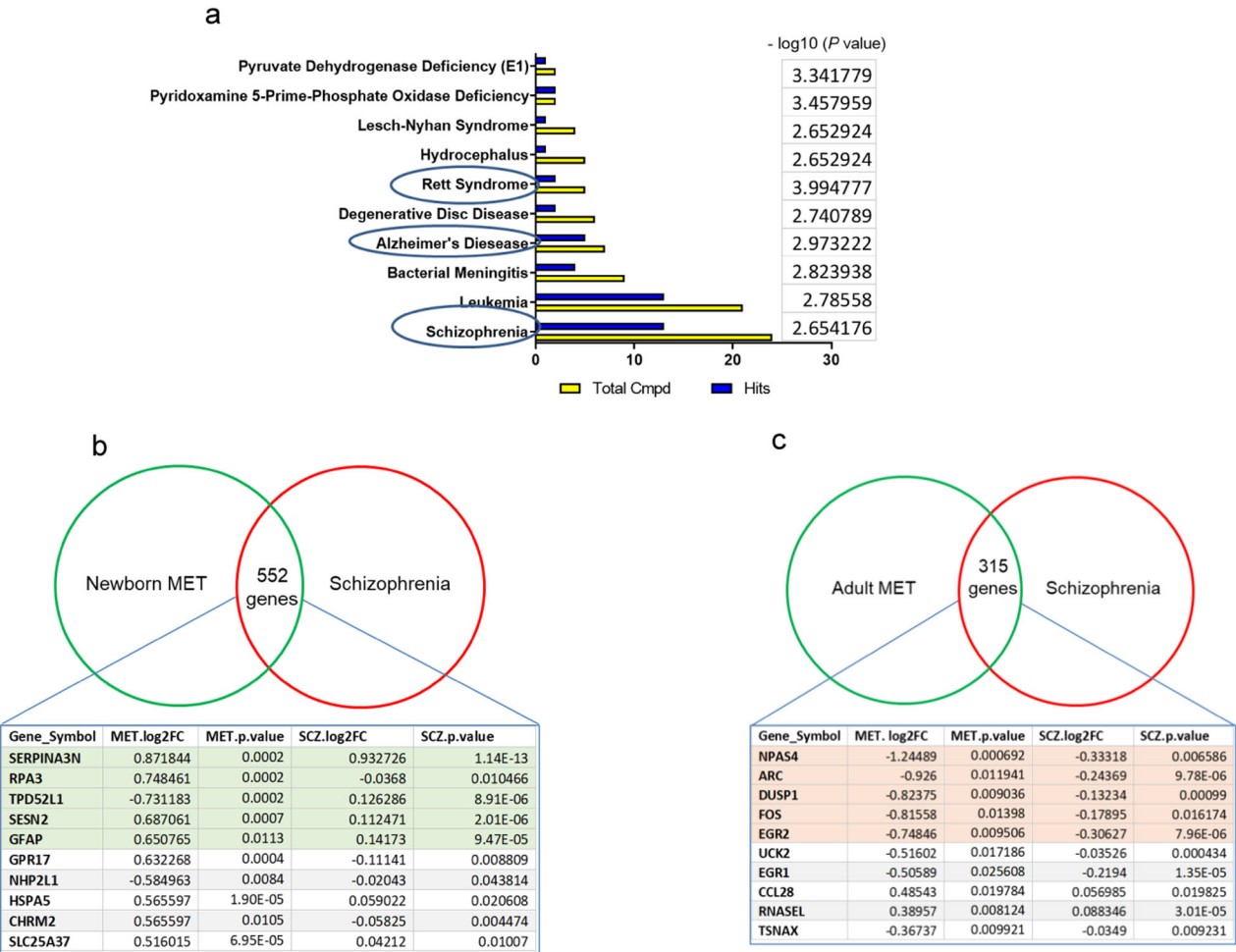

**Fig. 5 Integrated mouse-human transcriptomic analysis reveals overlap between genes whose expressions changed in MET mice and schizophrenia patients. a** Enrichment analysis of disease-associated metabolite sets in MET vs SAL brain. Diagram showing −log10(*p*) values for the top 10 associated diseases. **b** Genes whose expressions are changed in the brains of newborn MET mice and schizophrenia patients. **c** Genes whose expressions are changed in the brains of adult MET mice and schizophrenia patients.

phenotype observed in the offspring of mice injected with methionine during the last week of pregnancy is merely a result of the transfer of the symptoms from the mothers to their pups through mother–infant interaction. To answer this question, we assessed maternal behavior and examined the effects of crossed-fostering on the offspring behavioral phenotype. We demonstrated that MET mothers displayed normal maternal behaviors, evidenced by the normal pups' retrieval and nest building. We found that the behavioral deficits seen in MET offspring are not transferred through mothers' caregiving during the early life stage, as MET mothers did not transfer the phenotype to SAL-born pups. Rather, the behavioral deficits in MET mice seem to be a result of the in-utero changes. Moreover, the early mother-infant interaction during the first life stage did not rescue the behavioral deficits induced by prenatal methionine, as SAL mothers' caregiving was unable to reverse the hyperactivity, PPI deficits, and working memory impairments in MET pups.

The failure of SAL mothers to rescue the behavioral deficits in the MET-pups raises the question of which in-utero changes in the pups' brain metabolites are responsible for such long-lasting behavioral deficits induced by prenatal excessive methionine. The global brain metabolomics' analysis of 24 h-old pups reveals alterations in the MET pups' brain in the components of one-carbon metabolism including SAM, betaine, glycine, glutathione, methylglycinate, and serine. Given that exogenous methionine is

converted to SAM in the pregnant mice and their fetuses, the increase in SAM in the brains of the newborn MET pups' is not surprising. SAM is the main methyl donor in the body and is used in almost all methylation reactions. Methylation substrates can be DNA, RNA, proteins, phospholipids, carbohydrates, and neuro-transmitters. Therefore, excessive prenatal methionine would be expected to cause changes in all these substrates. Transcriptomic analyses reveal an increase in the expression of several genes encoding methylation enzymes that methylate DNA, mRNA, rRNA, and tRNA. MET brains also displayed increases in a number of methylated forms of molecules including amino acids (6-N-trimethyllysine), phospholipids (phosphatidylcholine), and the mitochondrial molecule Malonyl-carnitine.

Our metabolomic and transcriptomic results reveal profound dysregulations in lipid metabolism, including membrane and mitochondrial lipids. Lipids form over 50% of the brain's dry weight and they play crucial roles as the main brain's cell membrane structural components, as well as energy and signaling molecules. Membrane phospholipids are the key players of myelination, neuronal membrane geometrical shape and the physical characteristics, such as the fluidity and conductance. Therefore, the disruptions observed in the MET brains' membrane phospholipids (phosphatidylcholines, sphingomyelin, ceramides, and glyco-ceramides) would profoundly impact the functions of transmembrane receptors, ion channels, and other

membrane proteins. The neurotransmission signaling is also greatly impacted by abnormal lipid metabolism. Lysophosphatidylcholines (LPCs), byproducts of the phosphatidylcholine → arachidonic-acid hydrolysis, are potent membrane-dissolving molecules, and are known to induce demyelination[30–35], negatively impacting the speed and timing of neuronal signals. Therefore, the dysregulation of arachidonic acid signaling, along with the augmentation of LPC levels in MET pups, may imply decreased myelinations and, thus, disrupted circuit connectivity. Several pieces of evidence support the disrupted myelination in MET mice: 1) alongside the increase in LPCs, the myelination gene Gpr17 was upregulated. GPR17 is a G-protein coupled receptor that regulates oligodendrocyte differentiation in response to LPC-induced demyelination[30,36,37], and is overexpressed in brain tissues experiencing demyelination injuries[38–40]; 2) arachidonic acid derivatives (such as cysteinyl leukotrienes (LTC4 and LTD4), can directly regulate GPR17[41–43], and therefore the dysregulation of arachidonic acid pathways found in MET brains might be implicated in the demyelination; 3) the brains of newborn MET pups exhibited an increase in the mRNA and immunoreactivity of GFAP, a marker for neuroinflammation as well as myelination;[44–47] and 4) finally, the MET brains displayed an overexpression of Serpina3n, an astrocyte specific protein and neuro-inflammation marker. Strikingly, Serpin3a (the human analogue of Serpina3n) displays the highest fold-changes in the brains of schizophrenia patients (2-fold increase, $P = 1.14E-13$, FDR $= 1.62E-10$), and two-fold increase in autism ($P = 0.004154$, FDR $= 0.059$)[48]. Serpina3 is also overexpressed in Alzheimer's as it enhances amyloid beta fibrillization and accelerated plaque formation[49,50].

The mitochondrial lipid metabolism was also impacted by prenatal methionine. The metabolite that exhibited the highest change was malonyl-carnitine (increased 17-fold in the MET brains). Malonyl-carnitine concentrations reflect the levels of malonyl-CoA, a key regulator of fatty acid oxidation, which inhibits carnitine palmitoyltransferase 1 (Cpt1) and attenuates the mitochondrial uptake of long chain fatty acids[51–53]. Therefore, the accumulation of malonyl-carnitine in the MET pups' brains indicates a specific disruption of fatty-acid oxidation caused by impaired entry of long-chain acylcarnitine esters into the mitochondria. The decreases of mRNAs of Cpt1 and Mcat (mitochondrial malonylCoA:ACP acyltransferase) in the MET pups support an impaired synthesis and entry of long-chain acylcarnitines into the mitochondria. Alongside the disruption of mitochondrial lipid oxidation, metabolomic and transcriptomic analyses revealed dysregulated mitochondrial energy production. NAD+ dependent redox reactions were dysregulated, and 21 and 8 genes that are involved in NAD+ and glutathione pathways, respectively, were dysregulated in the MET brain.

MET pups displayed disruptions in the amino acid neurotransmitters' pathways and neurotransmitters' precursors, including decreased glutamine, glutamate, tryptophan, tyrosine, and increased aspartate. The decrease of glutamate was associated with the dysregulations of 17 genes involved in glutamate neurotransmission including glutamate transport, synthesis, release, receptors, and the regulation of NMDA receptors (Kcnma1). Kcnma1 displayed the highest increase in gene expression in MET brains (7-fold), and immunohistochemistry results revealed higher number of Kcnma1-expressing neurons in the hippocampus, thalamus and amygdala of MET brains. Kcnma1 encodes the large-conductance calcium-activated potassium channel, subfamily M, alpha member 1 (BK channel) pore-forming alpha subunit, which is involved in neuronal excitability and plasticity, and has been associated with neurological and psychiatric disorders, including epilepsy autism and Alzheimer's. While the activation of presynaptic BK channels decreases glutamate release[54–56], the activation of postsynaptic BK, which couples to GluN1, suppress NMDA-potentiated evoked excitatory postsynaptic potentials (EPSP)[57]. Our results suggest that early upregulation of Kcnma1 channels might provoke sustained hypoactivity of excitatory pathways in the hippocampus (reducing glutamate release, and suppressing NMDA receptors). In agreement, Arc expression was decreased in the brains of MET newborn pups. Interestingly, the MET pups exhibited an increase in the levels of glutamate-derived neurotransmitter, N-Acetylaspartylglutamate (NAAG), which acts as an agonist at glutamate metabotropic receptors 3 (mGluR3)[58,59] and has been associated with the pathogenesis of schizophrenia[60–64].

While our metabolomics and transcriptomic results support the phospholipid, mitochondria and glutamate dysfunction hypotheses of schizophrenia, these results point at similar changes occurring in schizophrenia and autism (Serpina3n, Gfap) as evidenced by the linkage analysis of the transcriptomics of the MET mice with the gene expression changes in schizophrenia and autism patients. The results, thus, suggest that the two neurodevelopmental disorders may partially share some etiological origins. The linkage results, however, should be cautiously interpreted, as the analysis of the intersections' size showed no significant difference from what can be expected by chance (the p values of the corresponding Fisher tests are close to 1).

Finally, our metabolomics and transcriptomic data reveal a number of previously unreported metabolites that can be used as biomarkers for the early detection of schizophrenia risk. These include malonyl-carnitine, LPCs, PCs, NAAG, ceramides, and arachidonic acid. The results also suggest previously unreported pathways and genes as therapeutic targets such as KCNMA1 inhibitors, GPR17 antagonists, SERPINA3N inhibitors, and selective PLA2G6 inhibitors.

While our study provides mechanistic insights into how prenatal disruption of the methionine metabolism produces behavioral deficits, particularly associated with inducing particular transcriptomic and metabolomics signatures, it poses some limitations. First, although methionine metabolism pathways are largely similar across different species, variabilities in the regulation of some enzymes involved in methionine metabolism between mice and human may raise questions concerning the degree to which our results from mouse models of schizophrenia mimic corresponding human disorder. Second, the direction of the changes in a few genes in MET newborn mice seems to be inconsistent with that in schizophrenia patients. This is not surprising, given that human schizophrenia transcriptomic analyses measure gene expression in the brains of adult humans, because schizophrenia symptoms appear only at late adolescence or adulthood. Thus, it is not possible to determine whether the same genes followed the same or opposite direction in expression changes in early stages of life of schizophrenia patients. Therefore, it is difficult to draw a concrete conclusion on whether the changes in the gene expressions in the newborn MET mice, which are in the opposite direction to those in schizophrenia patients, may play compensatory and/or protective mechanisms in patients' early life. Fourth, since our study did not directly connect the expression and metabolomics changes to the behaviors in the mice, within the same individual animals, we cannot conclude whether these metabolomics and transcriptomic abnormalities are responsible for the social and cognitive deficits seen in our mouse model. Sex differences may also play an important role and need to be investigated. These questions are important and they are the focus of our ongoing work including larger sample sizes.

In summary, our data indicate that excessive prenatal methionine plays a key role in promoting alterations in gene expression and brain metabolic pathways, associated with the cognitive and social impairments. Our data define early biomarkers that can reveal

neurodevelopmental defects, and identify targets for the protection/treatment of cognitive decline in schizophrenia and probably other psychiatric disorders, such as autism.

## Methods

**Animals, breeding procedures, and drug administration**. Swiss Webster mice, 8–9 weeks' age, were obtained from Charles River Laboratories (Wilmington, MA). One male and two female mice were group-housed, and pregnant mice were individually housed from the thirteen day of pregnancy (gestational day 13) until delivery. Pregnant mice that mated with the same male mouse were subcutaneously administered with L-Methionine (MET) at a dose of 750 mg/kg (Sigma-Aldrich) or saline (SAL) twice a day for 7–8 consecutive days from gestational day 14 until delivery. We chose a methionine dose and treatment schedule that were used previously by other groups[65–68], and then by our group[23,25], which corresponded to the regimens previously administered to humans, and was reported to exacerbate the psychotic symptoms in schizophrenia patients[26–29,69,70].

**Cross-fostering mouse model and animal care**. All experimental procedures were approved by the Institutional Animal Care and Use Committee of the University of California, Irvine and were performed in compliance with national and institutional guidelines for the care and use of laboratory animals. Pups of SAL and MET mice were raised by their biological mother (SAL/SAL and Met/MET) or were cross-fostered to a female of the other treatment (SAL/MET and MET/SAL for SAL pups with MET mother and MET pups with SAL mothers respectively) within 24 h of birth. Pups remained with their biological or foster mother until they were 21 days old, then male mice were selected for the study and group-housed in groups of 2–5 mice.

**Behavioral analyses**. *Maternal behavior: Nest building*. Nesting building behavior was assessed on PPD1 using the 0–5 scale of nest quality[71,72].

*Pup Retrieval*. The retrieval behavior test was performed at PPD2 to measure the mother's latency to retrieve the first pup and the total time required to retrieve 3 pups as described previously[71].

*Behavioral tests on the offspring*. Male mice were tested from postnatal week 8 to week 12 with a battery of behavioral paradigms in the following order: locomotion, spontaneous T-maze alternation, and prepulse inhibition. The sequence of specific assays spaced by 3–6 days inter-assay interval was adapted from our report[23].

*Locomotion activity*. Locomotor activity was measured as described previously[25]. Mice were allowed to habituate for 30 min in a locomotion test chamber (Med Associates, Inc.), and locomotor activity was recorded for 1 h and analyzed by Activity Monitor 5 software (Med Associates, Inc.).

*Spontaneous T-Maze alternation assay*. The spontaneous T-maze alternation assay was carried out as described previously[23]. Mice were placed in the start area of the T-maze (Accuscan Instruments, Inc.) and allowed 30 s of acclimation before the start of each trial. The door was then opened and mice were allowed to make a free choice into either side arm and to explore the arm for 30 s before being returned to the start area for the next trial. A total of eight trials were completed. Percentage of Alternation was calculated, and time to make a choice was recorded.

*Prepulse inhibition assay*. The Prepulse inhibition (PPI) assay was measured as previously described[73]. Mice were placed in the startle chambers for 5 min' acclimation with 65 dB background noise. The PPI session consisted of five different trials: no-stimulus trials, three prepulse trials and startle trials. No-stimulus trials consist of background noise only (65 dB). Startle trials consist of a 40 ms duration startle stimulus at 120 dB (p120). Prepulse trials consist of a 20 ms duration prepulse at 68 dB (pp3), 71 dB (pp6), or 77 dB (pp12), a 100 ms interstimulus interval, followed by a 40 ms duration startle stimulus at 120 dB. Test sessions began with 5 presentations of the p120 trial, followed by 10 presentations of the no-stimulus trial, p120 trials, pp3, pp6, and pp12 prepulse trials. The amount of PPI is calculated as a percentage score for each acoustic prepulse intensity.

**Brain metabolite analyses**. Metabolomics profiling on whole homogenate brain tissues collected from 24 h or 13 weeks' old mice was carried out at the West Coast Metabolomics center (University of California Davis). The Metabolomics profiles consisted of three platforms: (1) primary metabolites including carbohydrates and sugar phosphates, amino acids, hydroxyl acids, purines, pyrimidines, aromatics; (2) complex lipids; and (3) biogenic amines and methylated and acetylated amines. For the 24-h-old pups, tissues of one brain hemisphere (cortical-subcortical/mouse) were homogenized and extracted following the protocols first published in[74]. A differential analysis of the brain metabolites was performed using the Cyber-T program between the MET-treated and control groups for both newborn and adult mice groups, to identify the top up- or down-regulated metabolites[75,76].

**mRNA microarray analysis**. Microarray experiments and analysis were performed as previously described[77]. Total RNA was extracted (Qiagen) from one brain hemisphere of the 24-h-old pups (the opposite hemisphere to that used for metabolomics' analysis) according to the manufacturer's protocol, (the results of Transcriptomic analysis of the brains from adult Sal and MET pups were published in our previous report[23]. RNA was then reverse-transcribed into cDNA and analyzed by "whole-transcript transcriptomics" using the GeneAtlas microarray system (Affymetrix) and the manufacturer's protocols. The newly synthesized sscDNA was hybridized using MoGene 1.1 ST array strips (Affymetrix). Each array comprised 770,317 distinct 25-mer probes to probe an estimated 28,853 transcripts, with a median 27 probes per gene. Gene expression changes associated with the methionine treatment were analyzed with Transcriptome Analysis Console (TAC) software (Affymetrix) using Tukey's Bi-weight Average algorithm and default settings, generating fold-change and unpaired ANOVA values.

Top up- or down-regulated genes ($p < 0.05$) undergo further analyses including: (1) pathway enrichment analysis using the Pathway Common Database[78] and the ConsensusPathDB;[79] (2) identification of transcription factors (TFs) and RNA-binding proteins as important regulators; (3) transcription factor binding site (TFBS) enrichment analysis of all promoters using the MotifMap[80,81] database and the CHiPSeq database from ENCODE;[82] (4) analyses of all downstream targets of differentially expressed TFs and RBPs[80,81].

**Integrated metabolite-transcriptomic analysis**. The pathways and the enzyme-coding genes associated with each metabolite are identified using the Human Metabolome Database (HMDB)[83]. The protein-coding genes associated with the metabolites are matched to the transcriptomic data, linking the changes in gene expression levels to the changes in metabolite concentration levels. In addition, the functional enrichment analysis and disease pathway enrichment analysis of all metabolites are performed using the MetaboAnalyst web server[84].

**Integrated mouse-human transcriptomic analysis**. The differentially expressed genes in both adult and newborn mouse groups from the transcriptomic analysis are compared with differentially expressed genes in human autism and schizophrenia[48]. Common genes that are differentially expressed in both mouse and human autism, and both mouse and human schizophrenia are computed for adult and newborn mouse groups.

**Immunohistochemistry**. Twenty-four hours' pups were perfused transcardially with 0.9% saline and 4% paraformaldehyde. Brains were collected, post-fixed in 4% PFA for 24 h, and transferred to 30% sucrose at 4 °C. Fifty-micron sections were cut using a cryostat (Microm HM505E) and stored in phosphate buffered saline with 1% Sodium azide until immunostaining was performed. Three brain sections from four brains in each group were selected from regions at the following levels: hippocampus, amygdala, and somatosensory cortex, and thalamus. Sections were blocked with 4% goat serum in phosphate buffered saline (PBS) with 0.3% Triton x-100 for 60 min, and then incubated with the primary antibodies: rabbit anti Arc (1:500, abcam), GFAP (1:1000, Aves), rabbit anti KCNMA1 (1:200, Sigma-Aldrich) at room temperature for 24 h. Sections were washed and stained with goat anti rabbit Alexa-Fluor488 (1:500, Invitrogen) goat anti rabbit AlexaFluor555 (1:500 Invitrogen), and DAPI (1:10,000) to stain the nuclei. Sections were then mounted on slides with antifade solution. Leica Sp8 confocal microscope (UCI Optical Biology Core facility) was used for image acquisition and ImageJ and Volocity were used for analysis.

**Reporting summary**. Further information on research design is available in the Nature Research Reporting Summary linked to this article.

## Data availability

Our microarray and metabolomics-transcriptomics data have been deposited into GEO public repository: https://www.ncbi.nlm.nih.gov/geo/ (accession numbers: GSE153195 GSM4635464 GSM4635465 GSM4635466 GSM4635467 GSM4635468 GSM4635469 GSM4635470 GSM4635472 GSM4635473 GSM4635474 GSM4635475. The raw data of transcriptomic and metabolomic studies are available in the Supplementary Data 5. Other datasets generated during the current study can be requested from the corresponding author.

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

## Acknowledgements

We would like to thank Dr. Olivier Civelli for the helpful discussions. The work of A.A. was supported by the School of Medicine-UCI research grant. The work of S.C. and P.B. was supported in part by NIH grant GM123558 to P.B. The work of GWA was supported by NIH grant GM130377.

## Author contributions

A.A. designed the experiments, conducted behavioral experiments, and wrote the paper; W.A. and R.Y. conducted behavioral and immunostaining experiments, A.S. and G.W.A., conducted the transcriptomics analysis, S.C. and P.B. conducted the bioinformatics analysis

## Competing interests

The authors declare no competing interests.
