## [Peer Review File · Communications Biology]

Reviewers' comments:

Reviewer #1 (Remarks to the Author):

This study investigated the effects of excessive maternal exposure to methionine on metabolomic and transcriptomic signatures in offspring mice. The main findings were that excessive methionine exposure in utero led to significant changes in a variety of metabolic pathways, including one-carbon metabolism, neurotransmitters, mitochondrial function, and lipid metabolism. Many of the metabolic and gene expression alterations are seen in schizophrenia and autism. The authors also correlated these metabolic alterations with behavioral effects, including memory and prepulse inhibition, that were altered by the excessive methionine exposure. They as well determined that the effects of methionine exposure on the offspring were unrelated to maternal caregiving. The results of this study may identify therapeutic targets for schizophrenia and possibly autism.

Comments

1. Introduction (page 3, first paragraph): It would be helpful to the readers if reference to figure 2d is included here.

2. Introduction (page 3, second paragraph): Can the authors provide specific examples of the "constellation of behavioral, electrophysiological, and transcriptomic changes consistent with schizophrenia" that are caused by prenatal exposure to methionine?

3. Materials and Methods (page 5, second paragraph): Delete the last sentence on IACUC approval. It is redundant with the first sentence.

4. Materials and Methods (page 7, Brain Metabolite Analyses): The name of the place where the metabolomic profiles were conducted is the "West Coast Metabolomics Center" (not "West Coast Metabolics Center"). Also change the beginning of the first sentence to "Metabolomics profiling...", and the beginning of the second sentence to "Metabolomics profiles...". Also, just say "three platforms" instead of "three metabolics platforms".

5. Materials and Methods (page 8, line 2): Change "Tikey's" to "Tukey's".

6. Materials and Methods (page 9, immunohistochemistry): It is stated here that three brain sections were used for immunohistochemistry: striatum, hippocampus, and VTA. First, please spell out VTA – I presume it is the ventral tegmental area. Second, in the results section, different brain areas are described as being subjected to immunohistochemistry: amygdala, thalamus, and the somatosensory cortex. Please revise the methods accordingly.

7. Results (page 13-14): There seem to be some discrepancies between the text of the results sections and the figures. These are as follows:

- Mettl4 is listed in the text as associated with increased SAM, as indicated in Figure 4d. However, this gene does not appear in Table S8, nor does it appear in Figure 4a as connected to SAM. Please clarify.
- Figure 5b indicates 553 common genes, but the text says 552. Also, Figure 5b does not list the following genes mentioned in the text: Tpd52l1, Nhp2l1, and Nabp1. Please clarify.
- On page 13, 6 lines from the bottom, there is reference to a Figure b. Is this supposed to be Figure S2a? Please clarify.
- On page 13, last line, the gene Uck2 is listed as being one of the genes whose expression showed the largest fold change in Figure 5c. However, this gene is not listed in Figure 5c. Please clarify.
- On page 14, 2 lines from the bottom, there is reference to a Figure c. Is this supposed to be Figure

S2b? Please clarify.

Overall, it is not entirely clear how genes were chosen to be included in the figures and the text.

8. Discussion: Throughout the discussion the authors use the term "in uterus". This should be changed to "in utero" throughout.

Reviewer #2 (Remarks to the Author):

This paper examines metabolomics and transcriptional alterations following administration of prenatal excessive methionine (MET) on the brains and behavior of mice pups, while also investigating effects of cross-fostering. Alterations in mouse brains associated with prenatal MET were then compared with those observed in Schizophrenia and autism. This is an interesting paper and an extension of their previous work. There are several methodological details, including sample sizes, methods for testing overlap, and consideration of the direction of the changes observed across the disorders as well as for MET, that would be important to understand in order to gauge the strength of the findings and human relevance.

Summary

The main findings are 1) that behavioral phenotypes of excessive prenatal MET, which mimic several aspects of human schizophrenia, are independent of the early stage mother-infant interaction, and that caregiving by normal mothers is unable to rescue the behavioral deficits in the MET pups, and 2) prenatal MET exposure was associated with metabolomics and transcriptomic changes in glutamate transmission and lipid metabolism. These findings, especially the cross-fostering findings, are novel to my knowledge and are relevant to future studies in the field.

Study Design and Rationale

The study design was clearly laid out, however the rationale for the study design, including mouse strain choice, timing and dose of MET used in the study was not clear.

While Swiss Webster mice are likely a fine choice as a general mouse model, justification for using this strain should be given.

MET was administered in the 3rd /last week of pregnancy. It would be good to relate this to the comparable stage of human pregnancy. Further, further justification of selecting the end of pregnancy would have been helpful. Authors state that in their previous study "their final week of gestation at the time of brain development [23]", but because brain development occurs all the way throughout gestation and postnatally, this does not really justify this timing. More convincing would be an explanation of selection based on the timing of neurodevelopmental processes disrupted in Schizophrenia (and autism, which may not overlap). For example, neuronal migration, maturation, and differentiation abnormalities are found in autism, which occur throughout early and mid- gestation in humans.

It is unclear in this study why the dose of MET (750 mg/kg) twice a day was selected. Their previous paper explains that this is twice the normal dose. It seems prudent given this is a dose given in the previous study that elicited behavioral alterations. How does this relate to human doses? Is twice

the normal dose twice a day in humans feasible? (seems like it would be, but this should be stated to help with translation of findings to human health).

Why were only male offspring studied?

What are the sample sizes? For each stage of the study, it is unclear how many litters/mice were included (except it was given for the immunohistochemistry as 4 of each, which are not large numbers).

Methods

In the integrated mouse-human transcriptomic analysis, were tests of overlap in differentially expressed genes across disorders and MET (compared to expected overlap by chance) performed? Was there consideration of the direction of the up/down-regulation of genes associated with MET vs Schizophrenia/autism? Especially in evaluation of overlap in gene and metabolic lists? This is key to determining whether MET changes are increasing the disorder-related phenotypes, or whether they are protective against them.

Results

The results of the cross-fostering experiment are interesting and clearly indicate that alterations are due to the offspring's gestational MET exposure, rather than changes in maternal care postnatally, which has important clinical implications.

Results show alterations in lipid metabolism in the metabolomics analyses, and in the genes that are differentially expressed with MET. The consistency in changes across measures strengthens the findings. However, it is unclear whether the alterations are in the same direction, which is an important consideration.

The direction of the up/down-regulation of genes associated with MET vs Schizophrenia/autism is shown in supplemental tables and for the most part are consistent, with exceptions. For example, the number two gene (TPD52L1) overlapping between the newborn MET, autism, schizophrenia, while the p-values are all significant, the direction of the change for MET is counter to the that for ASD and Schizophrenia, which has important implications for treatment/prevention.

Discussion

The limitations of the study are not discussed. Discussion of how the results from this mouse model may not be translatable to humans, given many large differences in one-carbon metabolism between the two species, deserves mention, especially in light of the discussion of "novel metabolites that can be used as biomarkers for the early detection of schizophrenia risk... and novel pathways and genes as therapeutic targets."

The authors claim "our data indicate that excessive prenatal methionine plays a key role in promoting alterations in gene expression and brain metabolic pathways that drive the cognitive impairments in schizophrenia." Because the study really did not directly connect the expression and metabolomics changes to behaviors in the mice (within the same individual animals), this claim seems to take a bit

of a leap. If you can show that the mice with the most severe cognitive/behavior impairments also have the MET-induced brain expression and metabolite changes after sacrifice, then this would be supported by evidence.

Dear Editor in Chief,

We would like to thank you and the anonymous reviewers for reviewing our paper and for your valuable comments. We have made the requested changes, and hope that this improved manuscript is now acceptable to *Communications Biology*. We are attaching below our point-by-point response to the reviewers.

Reviewers' comments

Reviewer #1

This study investigated the effects of excessive maternal exposure to methionine on metabolomic and transcriptomic signatures in offspring mice. The main findings were that excessive methionine exposure in utero led to significant changes in a variety of metabolic pathways, including one-carbon metabolism, neurotransmitters, mitochondrial function, and lipid metabolism. Many of the metabolic and gene expression alterations are seen in schizophrenia and autism. The authors also correlated these metabolic alterations with behavioral effects, including memory and prepulse inhibition, that were altered by the excessive methionine exposure. They as well determined that the effects of methionine exposure on the offspring were unrelated to maternal caregiving. The results of this study may identify therapeutic targets for schizophrenia and possibly autism.

Comments

1. Introduction (page 3, first paragraph): It would be helpful to the readers if reference to figure 2d is included here.

Response: We thank the reviewer for this suggestion, and we now referred to figure 2d in the first paragraph of the introduction (page 3).

2. Introduction (page 3, second paragraph): Can the authors provide specific examples of the “constellation of behavioral, electrophysiological, and transcriptomic changes consistent with schizophrenia” that are caused by prenatal exposure to methionine?

Response: We have added the following sentences at the bottom of page 3: **“For example, MET mice displayed social deficits, cognitive impairments and augmented stereotypy, accompanied with decreased synaptic plasticity, and reduced local excitatory synaptic connections in the hippocampus. The schizophrenia-related genes Npas4, Arc, and Fos were downregulated in the MET mice”**.

3. Materials and Methods (page 5, second paragraph): Delete the last sentence on IACUC approval. It is redundant with the first sentence.

Response: We apologize for this repetition; we removed the last sentence from the paragraph (page 5).

4. Materials and Methods (page 7, Brain Metabolite Analyses): The name of the place where the metabolomic profiles were conducted is the “West Coast Metabolomics Center” (not “West Coast Metabolics Center”). Also change the beginning of the first sentence to “Metabolomics profiling...”, and the beginning of the second sentence to “Metabolomics profiles...”. Also, just say “three platforms” instead of “three metabolics platforms”.

Response: We thank the reviewer for this note. We made all the changes (page 7).

5. Materials and Methods (page 8, line 2): Change “Tikey’s” to “Tukey’s”.

Response: We made the correction (page 8, line 2)

6. Materials and Methods (page 9, immunohistochemistry): It is stated here that three brain sections were used for immunohistochemistry: striatum, hippocampus, and VTA. First, please spell out VTA – I presume it is the ventral tegmental area. Second, in the results section, different brain areas are described as being subjected to immunohistochemistry: amygdala, thalamus, and the somatosensory cortex. Please revise the methods accordingly.

Response: We thank the reviewer for this comment. What appears to be discrepant between the methods and the results in terms of the location of the immunohistochemistry, is explained by the fact that indeed some of the sections contain more than one region. For example: the hippocampus level also contains the amygdala, and the striatum level also contain somatosensory cortex. We have now removed the VTA. In order to avoid any confusion, we now use consistent regions levels in the methods and results (Page 9).

7. Results (page 13-14): There seem to be some discrepancies between the text of the results sections and the figures. These are as follows:

- *Mettl4* is listed in the text as associated with increased SAM, as indicated in Figure 4d. However, this gene does not appear in Table S8, nor does it appear in Figure 4a as connected to SAM. Please clarify.

Response: We made the corrections in Table S8 and Figure 4a by adding *mettl4*.

- Figure 5b indicates 553 common genes, but the text says 552. Also, Figure 5b does not list the following genes mentioned in the text: *Tpd52l1*, *Nhp2l1*, and *Nabp1*. Please clarify.

Response: We thank the reviewer for catching this mistake. We corrected the number in figure 5b to 552 (consistent with the text, and the list of the genes in table S10). We listed the genes Tpd52l1, Nhp2l1, and Nabp1 in the figure 5b.

- On page 13, 6 lines from the bottom, there is reference to a Figure b. Is this supposed to be Figure S2a? Please clarify.

Response: We apologize for any confusion. We fixed the figure number to Figure S2a.

- On page 13, last line, the gene Uck2 is listed as being one of the genes whose expression showed the largest fold change in Figure 5c. However, this gene is not listed in Figure 5c. Please clarify.

Response: We added Uck2 to the Figure 5c.

- On page 14, 2 lines from the bottom, there is reference to a Figure c. Is this supposed to be Figure S2b? Please clarify.

Response: We apologize for this mistake. We fixed the figure number to Fig S2b.

Overall, it is not entirely clear how genes were chosen to be included in the figures and the text.

Response: The top 10 genes that exhibited largest fold changes (in the MET pups) and were differentially expressed in schizophrenia patients were included in the text and figures. All other genes were listed in the Table S10.

8. Discussion: Throughout the discussion the authors use the term “in uterus”. This should be changed to “in utero” throughout.

Response: We have implemented this change throughout the discussion.

Reviewer #2

This paper examines metabolomics and transcriptional alterations following administration of prenatal excessive methionine (MET) on the brains and behavior of mice pups, while also investigating effects of cross-fostering. Alterations in mouse brains associated with prenatal MET were then compared with those observed in Schizophrenia and autism. This is an interesting paper and an extension of their previous work. There are several methodological details, including sample sizes, methods for testing overlap, and consideration of the direction of the changes observed across the disorders as well as for MET, that would be important to understand in order to gauge the strength of the findings and human relevance.

Summary

The main findings are 1) that behavioral phenotypes of excessive prenatal MET, which mimic several aspects of human schizophrenia, are independent of the early stage mother-infant interaction, and that caregiving by normal mothers is unable to rescue the behavioral deficits in the MET pups, and 2) prenatal MET exposure was associated with metabolomics and transcriptomic changes in glutamate transmission and lipid metabolism. These findings, especially the cross-fostering findings, are novel to my knowledge and are relevant to future studies in the field.

Response: We thank the reviewer for these positive remarks

Study Design and Rationale

- The study design was clearly laid out, however the rationale for the study design, including mouse strain choice, timing and dose of MET used in the study was not clear. While Swiss Webster mice are likely a fine choice as a general mouse model, justification for using this strain should be given.

Response: We thank the reviewer for this comment. We used Swiss Webster strains in our two previous studies [1, 2], which themselves followed previous studies [3-6] in particular those of Dr. Guidotti's group, which launched this model in male adult mice.

- MET was administered in the 3rd /last week of pregnancy. It would be good to relate this to the comparable stage of human pregnancy. Further, further justification of selecting the end of pregnancy would have been helpful. Authors state that in their previous study "their final week of gestation at the time of brain development [23]", but because brain development occurs all the way throughout gestation and postnatally, this does not really justify this timing. More convincing would be an explanation of selection based on the timing of neurodevelopmental processes disrupted in Schizophrenia (and autism, which may not overlap). For example, neuronal migration,

maturation, and differentiation abnormalities are found in autism, which occur throughout early and mid- gestation in humans.

Response: This is a very important comment, which lies at the core of the rationale for our study design. To address the comment, we have added the following sentences to the last paragraph of the introduction (page 4): “the timing of the MET administration in the third week of mouse pregnancy was selected to correspond to the comparable stage of human pregnancy. Given the alterations in methyl-donor requirements and methionine metabolism during the different gestation stages, a higher rate of transmethylation occurs in the 3rd trimester of gestation in pregnant women to meet the increase in methylation demands [7]. Further, maternal hyperhomocysteinemia (as a result of the disrupted methionine metabolism) during the third trimester of pregnancy has been associated with a twofold increase in the risk of schizophrenia in the adult offspring [8]”

- It is unclear in this study why the dose of MET (750 mg/kg) twice a day was selected. Their previous paper explains that this is twice the normal dose. It seems prudent given this is a dose given in the previous study that elicited behavioral alterations. How does this relate to human doses? Is twice the normal dose twice a day in humans feasible? (seems like it would be, but this should be stated to help with translation of findings to human health).

Response: We chose a methionine dose and treatment schedule that were used previously by other groups [3-6, 9], and then by our group [1, 2], which corresponded to the regimens previously administered to humans, and was reported to exacerbate the psychotic symptoms in schizophrenia patients [10-15].

- Why were only male offspring studied?

Response: This is a question that interests us as well. As we turn our attention to related sex studies in animal models of schizophrenia, we recognize that there is a wide variation in outcomes depending on the strain and behavioral models tested. Highlighting sex differences in a novel animal model will thus require paying rigorous attention to these details. In pursuit of this, we have been testing the female progeny in the behavioral models and have preliminarily seen the same results as in male mice. These studies are ongoing but we believe that the data on the males are in line with the vast majority of manuscripts describing behavioral analyses. Nevertheless, we have added a sentence to address this limitation in the discussion (page 20, lines 5-7): “**Sex differences may also play a significant role and need to be investigated. These questions are important and they are the focus of our ongoing work.**”

- What are the sample sizes? For each stage of the study, it is unclear how many litters/mice were included (except it was given for the immunohistochemistry as 4 of each, which are not large numbers).

Response: The exact numbers of each group in each experiment are detailed in the figure legend, please see page 25.

Methods

- In the integrated mouse-human transcriptomic analysis, were tests of overlap in differentially expressed genes across disorders and MET (compared to expected overlap by chance) performed?
- **Response:** The reviewer is raising an important point here and we agree with the reviewer that the intersections of gene lists can be analyzed on at least two different levels: (1) their sizes (in particular are they larger than what one would expect by pure chance?); and (2) their content in terms of enrichment of particular classes of genes and so forth. We did analyze the size of the intersections but we did not find them to be significantly different from what can be expected by chance (the p values of the corresponding Fisher tests are all close to 1). Thus, we focused on the content of the intersections and analyzed the top genes that are highly differential (with very low p-values) in both mice MET and human disorders.
- Was there consideration of the direction of the up/down-regulation of genes associated with MET vs Schizophrenia/autism? Especially in evaluation of overlap in gene and metabolic lists? This is key to determining whether MET changes are increasing the disorder-related phenotypes, or whether they are protective against them.
- **Response:** There was no specific consideration of the direction of the up/down-regulated genes but one can tell the direction from the tables by looking at the fold change of the gene or metabolite. Positive fold change indicates up-regulation and vice versa. However, to address this comment, we have added separated the up- and down-regulated genes in MET mice, together with their ranked fold change in human disorders. In this way, the reader can see more clearly if the up- or down-regulated genes in MET mice are up- or down-regulated in human disorders (please see Table S10).

Results

The results of the cross-fostering experiment are interesting and clearly indicate that alterations are due to the offspring's gestational MET exposure, rather than changes in maternal care postnatally, which has important clinical implications.

Response: we thank the reviewer for this positive remark.

- Results show alterations in lipid metabolism in the metabolomics analyses, and in the genes that are differentially expressed with MET. The consistency in changes across measures

strengthens the findings. However, it is unclear whether the alterations are in the same direction, which is an important consideration.

Response: We thank the reviewer for this comment; we have addressed the consistency in the lipid metabolism alteration in the results section. We have added the sentence: **“The increases in LPCs, LPE, fatty acids, and arachidonic acid were associated with overexpression of phospholipase enzymes such as Pla2g4b, Pla2g6, Plcxd2, Plcb3, and Napepld which hydrolyze phospholipids”**.

We would also like to point to our discussion, where we discuss the consistency between the metabolomics and transcriptomic results: “the accumulation of malonyl-carnitine in the MET pups’ brains indicates a specific disruption of fatty-acid oxidation caused by impaired entry of long-chain acylcarnitine esters into the mitochondria. The decreases of mRNAs of Cpt1 and Mcat (mitochondrial malonylCoA:ACP acyltransferase) in the MET pups support an impaired synthesis and entry of long-chain acylcarnitines into the mitochondria. Alongside the disruption of mitochondrial lipid oxidation, metabolomic and transcriptomic analyses revealed dysregulated mitochondrial energy production. NAD⁺ dependent redox reactions were dysregulated, and 21 and 8 genes that are involved in NAD⁺ and glutathione pathways, respectively, were dysregulated in the MET brain”

- The direction of the up/down-regulation of genes associated with MET vs Schizophrenia/autism is shown in supplemental tables and for the most part are consistent, with exceptions. For example, the number two gene (TPD52L1) overlapping between the newborn MET, autism, schizophrenia, while the p-values are all significant, the direction of the change for MET is counter to the that for ASD and Schizophrenia, which has important implications for treatment/prevention.

Response: This is a very important point, and we thank the reviewer for outlining it. We now address this point in the discussion as part of the limitations of our study as follows: **“the direction of the changes in a few genes in MET newborn mice seems to be inconsistent with that in schizophrenia patients. This is not surprising, given that human schizophrenia transcriptomic analyses measure gene expression in the brains of adult humans, because schizophrenia symptoms appear only at late adolescence or adulthood. Thus, it is not possible to determine whether the same genes followed the same or opposite direction in expression changes in early stages of life of schizophrenia patients. Therefore, it is difficult to draw a concrete conclusion on whether the changes in the gene expressions in the newborn MET mice, which are in the opposite direction to those in schizophrenia patients, may play compensatory and/or protective mechanisms in patients’ early life.”**

Discussion

- The limitations of the study are not discussed. Discussion of how the results from this mouse model may not be translatable to humans, given many large differences in one-carbon metabolism between the two species, deserves mention, especially in light of the discussion of “novel metabolites that can be used as biomarkers for the early detection of schizophrenia risk... and novel pathways and genes as therapeutic targets.”

Response: We have now added the following paragraph to the discussion to address the limitations of our study: **“While our study provides new mechanistic insights into how prenatal disruption of the methionine metabolism produces behavioral deficits, particularly associated with inducing particular transcriptomic and metabolomics signatures, it poses some limitations. First, although methionine metabolism pathways are largely similar across different species, variabilities in the regulation of some enzymes involved in methionine metabolism between mice and human may raise questions concerning the degree to which our results from mouse models of schizophrenia mimic corresponding human disorder. Second, the direction of the changes in a few genes in MET newborn mice seems to be inconsistent with that in schizophrenia patients. This is not surprising, given that human schizophrenia transcriptomic analyses measure gene expression in the brains of adult humans, because schizophrenia symptoms appear only at late adolescence or adulthood. Thus, it is not possible to determine whether the same genes followed the same or opposite direction in expression changes in early stages of life of schizophrenia patients. Therefore, it is difficult to draw a concrete conclusion on whether the changes in the gene expressions in the newborn MET mice, which are in the opposite direction to those in schizophrenia patients, may play compensatory and/or protective mechanisms in patients’ early life. Fourth, since our study did not directly connect the expression and metabolomics changes to the behaviors in the mice, within the same individual animals, we cannot conclude whether these metabolomics and transcriptomic abnormalities are responsible for the social and cognitive deficits seen in our mouse model. Sex differences may also play a significant role and need to be investigated. These questions are important and they are the focus of our ongoing work.”**

- The authors claim “our data indicate that excessive prenatal methionine plays a key role in promoting alterations in gene expression and brain metabolic pathways that drive the cognitive impairments in schizophrenia.” Because the study really did not directly connect the expression and metabolomics changes to behaviors in the mice (within the same individual animals), this claim seems to take a bit of a leap. If you can show that the mice with the most severe cognitive/behavior impairments also have the MET-induced brain expression and metabolite changes after sacrifice, then this would be supported by evidence.

Response: We absolutely agree with the reviewer. Since we did not conduct a correlation study within individual animals, we have removed the following sentence from the last paragraph of the discussion: “our data indicate that excessive prenatal methionine plays a key role in promoting alterations in gene expression and brain metabolic pathways that drive the cognitive impairments in schizophrenia.” Instead, we have replaced it with the following statement: **“our data indicate that excessive prenatal methionine plays a key role in promoting alterations in gene expression and brain metabolic pathways, associated with the cognitive and social impairments”**

References

1. Alachkar, A., et al., *Prenatal one-carbon metabolism dysregulation programs schizophrenia-like deficits*. Mol Psychiatry, 2018. **23**(2): p. 282-294.

2. Wang, L., et al., *A Methionine-Induced Animal Model of Schizophrenia: Face and Predictive Validity*. Int J Neuropsychopharmacol, 2015. **18**(12).
3. Tremolizzo, L., et al., *An epigenetic mouse model for molecular and behavioral neuropathologies related to schizophrenia vulnerability*. Proc Natl Acad Sci U S A, 2002. **99**(26): p. 17095-100.
4. Dong, E., et al., *Clozapine and sulpiride but not haloperidol or olanzapine activate brain DNA demethylation*. Proc Natl Acad Sci U S A, 2008. **105**(36): p. 13614-9.
5. Tueting, P., et al., *L-methionine decreases dendritic spine density in mouse frontal cortex*. Neuroreport, 2010. **21**(8): p. 543-8.
6. Matrisciano, F., et al., *Activation of group II metabotropic glutamate receptors promotes DNA demethylation in the mouse brain*. Mol Pharmacol, 2011. **80**(1): p. 174-82.
7. Dasarathy, J., et al., *Methionine metabolism in human pregnancy*. Am J Clin Nutr, 2010. **91**(2): p. 357-65.
8. Brown, A.S., et al., *Elevated prenatal homocysteine levels as a risk factor for schizophrenia*. Arch Gen Psychiatry, 2007. **64**(1): p. 31-9.
9. !!! INVALID CITATION !!!
10. Pollin, W., P.V. Cardon, Jr., and S.S. Kety, *Effects of amino acid feedings in schizophrenic patients treated with iproniazid*. Science, 1961. **133**(3446): p. 104-5.
11. Antun, F.T., et al., *The effects of L-methionine (without MAOI) in schizophrenia*. J Psychiatr Res, 1971. **8**(2): p. 63-71.
12. Brune, G.G. and H.E. Himwich, *Effects of methionine loading on the behavior of schizophrenic patients*. J Nerv Ment Dis, 1962. **134**: p. 447-50.
13. Cohen, S.M., et al., *The administration of methionine to chronic schizophrenic patients: a review of ten studies*. Biol Psychiatry, 1974. **8**(2): p. 209-25.
14. Spaide, J., et al., *Methionine and tryptophan loading in schizophrenic patients receiving a MAO inhibitor: correlation of behavioral and biochemical changes*. Biol Psychiatry, 1969. **1**(3): p. 227-33.
15. Ananth, J.V., et al., *Nicotinic acid in the prevention and treatment of methionine-induced exacerbation of psychopathology in schizophrenics*. Can Psychiatr Assoc J, 1970. **15**(1): p. 15-20.

Figure 4a (we added METTL4)

Figure 5b (We corrected the number of genes to 552, and added the missing genes to the list)

Gene_Symbol	MET.log2FC	MET.p.value	SCZ.log2FC	SCZ.p.value
SERPINA3N	0.871844	0.0002	0.932726	1.14E-13
RPA3	0.748461	0.0002	-0.0368	0.010466
TPD52L1	-0.731183	0.0002	0.126286	8.91E-06
SESN2	0.687061	0.0007	0.112471	2.01E-06
GFAP	0.650765	0.0113	0.14173	9.47E-05
GPR17	0.632268	0.0004	-0.11141	0.008809
NHP2L1	-0.584963	0.0084	-0.02043	0.043814
HSPA5	0.565597	1.90E-05	0.059022	0.020608
CHRM2	0.565597	0.0105	-0.05825	0.004474
SLC25A37	0.516015	6.95E-05	0.04212	0.01007

Figure 5c (we added the missing genes to the list (Uck2))

Gene_Symbol	MET. log2FC	MET.p.value	SCZ.log2FC	SCZ.p.value
NPAS4	-1.24489	0.000692	-0.33318	0.006586
ARC	-0.926	0.011941	-0.24369	9.78E-06
DUSP1	-0.82375	0.009036	-0.13234	0.00099
FOS	-0.81558	0.01398	-0.17895	0.016174
EGR2	-0.74846	0.009506	-0.30627	7.96E-06
UCK2	-0.51602	0.017186	-0.03526	0.000434
EGR1	-0.50589	0.025608	-0.2194	1.35E-05
CCL28	0.48543	0.019784	0.056985	0.019825
RNASEL	0.38957	0.008124	0.088346	3.01E-05
TSNAX	-0.36737	0.009921	-0.0349	0.009231

REVIEWERS' COMMENTS:

Reviewer #2 (Remarks to the Author):

We chose a methionine dose and treatment schedule that were used previously by other groups [3-6, 9], and then by our group [1, 2], which corresponded to the regimens previously administered to humans, and was reported to exacerbate the psychotic symptoms in schizophrenia patients [10-15]. This information should be added to the paper.

We did analyze the size of the intersections but we did not find them to be significantly different from what can be expected by chance (the p values of the corresponding Fisher tests are all close to 1). This is not supportive evidence and should be pointed out in the discussion.

The sample sizes are small. This should be added to the limitations.

Replication would strengthen confidence in the findings.

Dear Editor in Chief,

We would like to thank you and the reviewers for reviewing our paper and for your valuable comments. We have made the requested changes, and hope that this improved manuscript is now acceptable to *Communications Biology*. We are attaching below our point-by-point response to the reviewers.

Reviewers' comments

Reviewer #2 (Remarks to the Author):

1- We chose a methionine dose and treatment schedule that were used previously by other groups [1-4], and then by our group [5, 6] which corresponded to the regimens previously administered to humans, and was reported to exacerbate the psychotic symptoms in schizophrenia patients [7-12].

This information should be added to the paper.

Response: We now added the sentences to methods section (Page 16, line 8).

2- We did analyze the size of the intersections but we did not find them to be significantly different from what can be expected by chance (the p values of the corresponding Fisher tests are all close to 1). This is not supportive evidence and should be pointed out in the discussion.

Response: We now point out to this in the discussion (page 13 last line, and page 14 the first two lines)

3- The sample sizes are small. This should be added to the limitations.

Response: We address the sample size (page 15, line 3)

4- Replication would strengthen confidence in the findings.

Response: We absolutely agree with the reviewer. With regard to the behavioral tests, we confirm that the experiments were replicated several times producing the same results. That is evidenced by obtaining the same behavioral phenotypes induced by excessive prenatal methionine in our previous study and the current study [5].

1. Tremolizzo, L., et al., *An epigenetic mouse model for molecular and behavioral neuropathologies related to schizophrenia vulnerability*. Proc Natl Acad Sci U S A, 2002. **99**(26): p. 17095-100.
2. Dong, E., et al., Clozapine and sulpiride but not haloperidol or olanzapine activate brain DNA demethylation. Proc Natl Acad Sci U S A, 2008. 105(36): p. 13614-9.
3. Tueting, P., et al., L-methionine decreases dendritic spine density in mouse frontal cortex. Neuroreport, 2010. 21(8): p. 543-8.
4. Matrisciano, F., et al., Activation of group II metabotropic glutamate receptors promotes DNA demethylation in the mouse brain. Mol Pharmacol, 2011. 80(1): p. 174-82.
5. Alachkar, A., et al., Prenatal one-carbon metabolism dysregulation programs schizophrenia-like deficits. Mol Psychiatry, 2018. 23(2): p. 282-294.

6. Wang, L., et al., A Methionine-Induced Animal Model of Schizophrenia: Face and Predictive Validity. *Int J Neuropsychopharmacol*, 2015. 18(12).
7. Pollin, W., P.V. Cardon, Jr., and S.S. Kety, Effects of amino acid feedings in schizophrenic patients treated with iproniazid. *Science*, 1961. 133(3446): p. 104-5.
8. Antun, F.T., et al., The effects of L-methionine (without MAOI) in schizophrenia. *J Psychiatr Res*, 1971. 8(2): p. 63-71.
9. Brune, G.G. and H.E. Himwich, Effects of methionine loading on the behavior of schizophrenic patients. *J Nerv Ment Dis*, 1962. 134: p. 447-50.
10. Cohen, S.M., et al., The administration of methionine to chronic schizophrenic patients: a review of ten studies. *Biol Psychiatry*, 1974. 8(2): p. 209-25.
11. Spaide, J., et al., Methionine and tryptophan loading in schizophrenic patients receiving a MAO inhibitor: correlation of behavioral and biochemical changes. *Biol Psychiatry*, 1969. 1(3): p. 227-33.
12. Ananth, J.V., et al., Nicotinic acid in the prevention and treatment of methionine-induced exacerbation of psychopathology in schizophrenics. *Can Psychiatr Assoc J*, 1970. 15(1): p. 15-20.